# *Toxoplasma gondii* UBL-UBA Shuttle Protein DSK2s Are Important for Parasite Intracellular Replication

**DOI:** 10.3390/ijms22157943

**Published:** 2021-07-26

**Authors:** Heng Zhang, Xu Yang, Zhu Ying, Jing Liu, Qun Liu

**Affiliations:** 1National Animal Protozoa Laboratory, College of Veterinary Medicine, China Agricultural University, Beijing 100193, China; zhrealm@163.com (H.Z.); 15801295256@163.com (X.Y.); YZyingzhu@163.com (Z.Y.); 2Key Laboratory of Animal Epidemiology of the Ministry of Agriculture, College of Veterinary Medicine, China Agricultural University, Beijing 100193, China

**Keywords:** *Toxoplasma gondii*, UBL-UBA shuttle protein, degradation of ubiquitinated proteins, cell division

## Abstract

*Toxoplasma gondii* (*T. gondii*) is an important human and veterinary pathogen causing life-threatening disease in immunocompromised patients. The UBL-UBA shuttle protein family are important components of the ubiquitin–proteasome system. Here, we identified a novel UBL-UBA shuttle protein DSK2b that is charactered by an N-terminal ubiquitin-like domain (UBL) and a C-terminal ubiquitin-associated domain (UBA). DSK2b was localized in the cytoplasm and nucleus. The deletion of *dsk2b* did not affect the degradation of ubiquitinated proteins, parasite growth in vitro or virulence in mice. The double-gene knockout of *dsk2b* and its paralogs *dsk2a* (ΔΔ*dsk2adsk2b*) results in a significant accumulation of ubiquitinated proteins and the asynchronous division of *T. gondii*. The growth of ΔΔ*dsk2adsk2b* was significantly inhibited in vitro, while virulence in mice was not attenuated. In addition, autophagy occurred in the ΔΔ*dsk2adsk2b,* which was speculated to degrade the accumulated ubiquitinated proteins in the parasites. Overall, DSK2b is a novel UBL-UBA shuttle protein contributing to the degradation of ubiquitinated proteins and is important for the synchronous cell division of *T. gondii.*

## 1. Introduction

*Toxoplasma gondii* is a common veterinary and human pathogen that typically lives as an obligate intracellular parasite within host cells [1]. The host infected by *T. gondii* is usually asymptomatic; however, the parasite is life-threatening in immunocompromised individuals. Its fast growth is responsible for the pathology that develops in toxoplasmosis patients [2].

Ubiquitination is an important post-translational modification that regulates many cellular processes such as protein trafficking, protein localization, cellular signaling and transcription [3]. The most important function of ubiquitination is to mediate protein degradation by the proteasome, which is named the ubiquitin–proteasome system (UPS) [4]. Approximately 80–90% of ubiquitinated proteins are degraded by the UPS in a eukaryotic cell [5].

The ubiquitin proteome reveals that ubiquitination is ubiquitous in *T. gondii* and is implicated as a critical regulator of cell division and cell-cycle transitions [6]. The proteasome is highly conserved in *T. gondii* [7]. These studies suggest that the UPS exists and plays important roles in *T. gondii*.

UBL-UBA shuttle proteins are important components of the UPS [8]. The UBL-UBA shuttle proteins are charactered by an N-terminal ubiquitin-like domain (UBL) and at least a C-terminal ubiquitin-associated domain (UBA). Recent studies showed that UBL-UBA shuttle proteins bind ubiquitinated proteins via the UBA domain and interact with the proteasome via the UBL domain [9,10], which promotes or inhibits the degradation of proteins [11].

DDI1 (DNA damage inducible protein 1), RAD23 (UV excision repair protein rad23 protein) and DSK2 (a ubiquitin family protein) are the most studied UBL-UBA shuttle proteins. DDI1 is essential for the degradation of HO endonuclease, which initiates a mating-type switch by generating a double-strand break at the mating-type locus in yeast [12]. RAD23 and DSK2 were reported to participate in endoplasmic-reticulum-associated degradation and the OLE-1 gene induction pathway [13]. In addition, the UBL-UBA shuttle protein family is not only involved in the degradation of ubiquitinated proteins but also plays important roles in DNA repair [14], spindle pole body duplication [15] and protein secretion [16]. Thus, the function of the UBL-UBA shuttle protein family is intriguing.

In this study, we identified a novel UBL-UBA shuttle protein (TGME49_240700) in *T. gondii*. Phylogenetic analysis indicated that TGME49_240700 is homologous to DSK2, and thus we named it ‘DSK2b’ and named the previous identified DSK2 ‘DSK2a.’ DSK2b contains an N-terminal UBL domain and a C-terminal UBA domain. The ectopic expression of HA-DSK2b showed that DSK2b was localized in the cytoplasm and nucleus of *T. gondii*. The deletion of *dsk2b* did not affect the degradation of ubiquitinated proteins, parasite growth or virulence in mice. The double-gene knockout of *dsk2a* and *dsk2b* resulted in a significant accumulation of ubiquitinated proteins. The accumulation of ubiquitinated proteins caused asynchronous replication and growth inhibition of *T. gondii*. In addition, ATG8-PE was increased in the DSK2s’ mutant strain, suggesting that autophagy occurred.

## 2. Results

### 2.1. DSK2b Is a Novel UBL-UBA Shuttle Protein in T. gondii

A UBL-UBA shuttle protein is charactered by a N-terminal ubiquitin-like domain (UBL) and a C-terminal ubiquitin-associated domain (UBA). Our previous study identified three UBL-UBA shuttle proteins (DDI1, RAD23 and DSK2) in *T. gondii*. To identify new shuttle proteins, BLAST using ubiquitin as a query sequence was performed to identify proteins containing a UBL domain. The results showed that thirteen proteins are homologous to ubiquitin. Further research based upon the SMART database showed that four of the thirteen proteins contain a UBL domain and a UBA domain (Figure 1a), including DDI1, RAD23, DSK2 and TGME49_240700. Phylogenetic analysis showed that TGME49_240700 was homologous to DSK2 (Figure 1b). Domain analysis showed that TGME49_240700 contained a typical UBL domain (residues 78–146) and a C-terminal UBA domain (residues 561–600). Both DSK2 and TGME49_240700 contain a CBM heat shock chaperonin-binding domain (HSCB) (Figure 1c). The predicted three-dimensional (3D) structures of DSK2b showed that several β sheets and an α-helix constitute the classical UBL domain, and a bundle of three α-helices constitute the classical UBA domain (Figure 1d). Thus, TGME49_240700 is a novel UBL-UBA shuttle protein, and has been named ‘DSK2b.’ At the same time, we named the previously identified DSK2 ‘DSK2a.’

### 2.2. DSK2b Is Localized in the Cytoplasm and Nucleus of T. gondii

To determine the localization of DSK2b in *T. gondii*, we first attempted to tag the *dsk2b* gene with three epitope HA tags at the C-terminal of the genomic locus. However, several attempts failed. We guessed that the HA tags fused to the C-terminal were undetectable. Thus, we constructed an ectopic expression strain, in which the 3× HA tags were fused to the N-terminal of the *dsk2b* coding sequence. The transcription of HA-DSK2b was initiated by a GRA1 promoter (Figure 2a). Western blotting showed that HA-DSK2b was successfully expressed in the parasites (Figure 2b). An IFA using an HA monoclonal antibody showed that HA-DSK2b was localized in the cytoplasm and nucleus of *T. gondii*. (Figure 2c). The ectopic expression of HA-DSk2b in Δ*ku80* make it an overexpression strain. A plaque assay showed that the overexpression of DSK2b did not affect the growth of *T. gondii* (Figure 2d,e).

### 2.3. Deletion of DSK2b Did Not Affect the Degradation of Ubiquitinated Proteins and Parasite Growth

To address the function of DSK2b, we constructed the gene knockout strain of *dsk2b* (Δ*dsk2b*) by CRISPR/Cas9-mediated homologous recombination (Figure 3a). Diagnostic PCR showed that the *dsk2b* gene was knocked out successfully (Figure 3b).

The function of a UBL-UBA shuttle protein is to transfer ubiquitinated proteins to the proteasome for degradation. Thus, we detected the K48-linked polyubiquitin chains (a modification associated with directing ubiquitin-mediated protein turnover) in Δ*dsk2b* by Western blotting. The results showed that the ubiquitinated proteins in Δ*dsk2b* were not affected (Figure 3c,d). An invasion assay, replication assay and plaque assay showed that the deletion of *dsk2b* did not affect the invasion and growth of *T. gondii* (Figure 3e–h). A virulence assay in mice showed that *dsk2b* was dispensable for *T. gondii* pathogenesis (Figure 3i).

### 2.4. Deletion of DSK2s Results in the Accumulation of Ubiquitinated Proteins

DSK2b and DSK2a are paralogs and thus their function may be complementary. Therefore, we constructed a double-gene knockout strain (ΔΔ*dsk2adsk2b*) by CRISPR/Cas9-mediated homologous recombination (Figure 4a). Diagnostic PCR showed that both genes were knocked out successfully (Figure 4b). Western blotting showed that ubiquitinated proteins were significantly accumulated in the ΔΔ*dsk2adsk2b* (Figure 4c,d). Our previous study showed that the accumulation of ubiquitinated proteins in Δ*dsk2a* was also not affected [17]. These results suggest that the functions of DSK2b and DSK2a are complementary.

### 2.5. Deletion of DSK2s Results in Asynchronous Replication and Inhibits T. gondii Growth

Ubiquitination plays important roles during cell-cycle transitions in *T. gondii* [6]. To determine the effect of the accumulated ubiquitinated proteins on *T. gondii*, we performed an IFA to observe the division of ΔΔ*dsk2adsk2b*. The inner membrane complex protein 1 (IMC1) is a frequently-used marker for *T. gondii* replication [18]. IFA showed that the division of Δ*ku80*, Δ*dsk2a* and Δ*dsk2b* in the same parasitophorous vacuole (PV) was synchronous; however, the division of ΔΔ*dsk2adsk2b* in the same PV was asynchronous (Figure 5a). The duplication of the centrosome occurs before daughter IMC formation [19]. Subsequently, we detected the division of ΔΔ*dsk2adsk2b* using IMC1 and centrin1 (a protein localized on the centrosome used to mark the replication of *T. gondii*). The results showed that the duplication of centrosome in some ΔΔ*dsk2adsk2b* tachyzoites was before IMC formation and was asynchronous, characterized by a variable number of centrosomes in different tachyzoites (Figure 5b). The duplication of the centrosome is associated with apicoplast during *T. gondii* endodyogeny [20]. IFA showed that more than two apicoplasts were observed in some tachyzoites of ΔΔ*dsk2adsk2b* during division (Figure 5c). These results suggest that DSK2s are important for the normal division of *T. gondii*.

Subsequently, we determined the effect of the DSK2s’ deletion on parasite growth. A replication assay showed that most PVs contain 8 or 16 tachyzoites in the Δ*ku80* strain. However, most PVs in ΔΔ*dsk2adsk2b* contain 4 or 8 tachyzoites and about 20% of PVs contain abnormal numbers of tachyzoites (non-2^n^ numbers) due to asynchronous replication (Figure 6a). A plaque assay showed that the plaque size of ΔΔ*dsk2adsk2b* was significantly decreased (Figure 6b,c). A virulence assay showed that mice infected with ΔΔ*dsk2adsk2b* survived more than two days longer than those with the parental strain (Figure 6d). These results suggest that the deletion of the DSK2s inhibits the growth of *T. gondii* in vitro; however, it did not affect virulence in mice.

### 2.6. Autophagy Pathway Was Occurred in DSK2s Mutant Strain

The UPS and autophagy are two major quality control systems responsible for the degradation of proteins and organelles in eukaryotic cells [4]. DSK2s were not only a UBL-UBA shuttle protein in the UPS, but also autophagic cargo receptors [21]. If the deletion of *dsk2s* interfered with the UPS, then we want to determine its effect on autophagy. ATG8 is an autophagosome marker in *T. gondii* and increased ATG8-PE (lapidated ATG8) marks autophagy [22]. Western blotting showed that the rapidly migrated ATG8-PE was increased in the ΔΔ*dsk2adsk2b* strain (Figure 7a). The aggregated ATG8 was observed in the ΔΔ*dsk2adsk2b* strain rather than the parental Δ*ku80* strain using IFA (Figure 7b). These results suggested that autophagy occurred in the ΔΔ*dsk2adsk2b* strain. Autophagy is another pathway for degrading ubiquitinated proteins. Thus, we speculated that the observed autophagy was a result of the degradation of the accumulated ubiquitinated proteins in ΔΔ*dsk2adsk2b*.

## 3. Discussion

Ubiquitination is an important post-translation modification regulating many cellular processes. The most studied function of ubiquitination is its role in the degradation of ubiquitinated proteins mediated by the ubiquitin–proteasome system. UBL-UBA shuttle proteins are important components of the ubiquitin–proteasome system. Here, we identified a novel UBL-UBA shuttle protein, named DSK2b, in *T. gondii*. DSK2b localized in the cytoplasm and nucleus of *T. gondii*. By constructing a gene knockout strain of *dsk2b*, we revealed that DSK2b was not essential for the degradation of ubiquitinated proteins in *T. gondii*. However, when *dsk2b* was knocked out with its paralog *dsk2a*, a significant accumulation of ubiquitinated proteins was observed in the parasites. Double-gene knockout of *dsk2s* results in the asynchronous division of *T. gondii* and inhibits the growth of the parasite. In addition, the deletion of *dsk2s* results in autophagy.

Failed tagging at the C-terminus of *dsk2b* was similar to our previous research on other UBL-UBA shuttle proteins. Thus, we speculated that tags fused to the shuttle proteins C-terminus may be undetectable or are cut off during protein maturation, which might explain why the tags of the yeast shuttle proteins were located at the N-terminus [23].

Ubiquitination is an abundant post-translational modification in *T. gondii*. The replication of *T. gondii* is a complex process, implicating many ubiquitinated proteins [6]. Our previous studies have revealed that the mutation of three other UBL-UBA shuttle proteins result in ubiquitinated protein accumulation and the asynchronous replication of *T. gondii* [17]. This study also showed that the accumulated ubiquitinated proteins cause asynchronous division of ΔΔ*dsk2adsk2b.* Thus, we speculated that the substrates of UBL-UBA shuttle proteins are extensive and some of them may be important regulators of *T. gondii* replication. The deletion of shuttle proteins may result in the aberrant degradation of some cell-cycle-related ubiquitinated proteins and interfere with division.

Eukaryotic cells have feedback regulation mechanisms or compensatory pathways to maintain the cellular environmental homeostasis. Excepting the UPS, autophagy is another pathway to degrade ubiquitinated proteins [24]. *T. gondii* is capable of autophagy and the process can be induced by stress conditions or to maintain cell homeostasis during normal growth and development [22]. We speculated that the accumulated ubiquitinated proteins in ΔΔ*dsk2adsk2b* may be a stress, which causes autophagy to degrade the accumulated ubiquitinated proteins. However, the ubiquitinated proteins were still accumulated in ΔΔ*dsk2adsk2b*, suggesting the autophagy cannot replace the function of UPS completely.

DSK2 was studied extensively in yeast and *Arabidopsis thaliana* [15,21]. Yeast DSK2 is not essential for yeast survival [25], but it is important for salt tolerance [26]. *Beauveria bassiana* DSK2 is involved in multi-stress tolerance and thermal adaptation ability [27]. In our studies, although the ΔΔ*dsk2adsk2b* grew slowly, it still can be passaged stably and was virulent to mice. Thus, we speculated that *T. gondii* lacking DSK2s maybe sensitive to some undiscovered stress stimulation. In addition, the CBM heat-shock chaperonin-binding domain (HSCB) distinguishes DSK2 from other UBL-UBA shuttle proteins. Whether the HSCB is related to stress stimulation would make an intriguing study.

We observed several phenotypic changes in the ΔΔ*dsk2adsk2b* strain. Although the phenotypes were not verified by constructing a complementary strain, we determined the phenotype by performing assays in four different clones. The results of each clone are reproducible, suggesting the phenotypic changes are indeed caused by *dsk2s* deletion.

Overall, as the essential components of UPS, UBL-UBA shuttle proteins may regulate many cellular processes by contributing to the degradation of ubiquitinated proteins.

## 4. Materials and Methods

### 4.1. Parasites, Host Cells and Antibodies

The RHΔ*ku80* strain was used as the parental parasite. The Δ*dsk2a* strain (the RHΔ*ku80* strain lacking the *dsk2a* gene) were also used. These two strains were preserved in our laboratory.

Parasites were cultured in human foreskin fibroblast cells (HFFs) (ATCC, Rockefeller, MD, USA) growing in Dulbecco’s Modified Eagle’s Medium (DMEM) supplemented with 10% fetal bovine serum (FBS) at 37 °C in 5% CO_2_.

Primary antibodies used in the study: rabbit anti-ubiquitin monoclonal Ab (ab134953) was purchased from abcam (Cambridge, UK), mouse anti-HA MAb was purchased from Sigma (St. Louis, MO, USA), and mouse anti-IMC1, anti-ATG8 and anti-actin, and rabbit anti-GAP45, anti-SAG1, anti-ENR and anti-centrin1 polyclonal antibodies were preserved in our laboratory.

Secondary antibodies used in the study: FITC-conjugated goat anti-mouse IgG (H + L) and goat anti-rabbit IgG (H + L), Cy3-conjugated goat anti-rabbit IgG (H + L) and goat anti-mouse IgG (H + L) horseradish peroxidase (HRP) were purchased from Sigma (St. Louis, MO, USA).

### 4.2. Immunofluorescence Assays and Western Blotting

Immunofluorescence assays (IFAs) were carried out as described previously [28]. Briefly, confluent HFFs on glass coverslips were infected with parasites for an appropriate time before fixing with 4% formaldehyde for 20 min. Then the cells were permeabilized with 0.25% Triton X-100 for 20 min and blocked with 3% bovine serum albumin (BSA) in phosphate-buffered saline (PBS) for 30 min. Primary antibody diluted in 3% BSA–PBS was incubated for 1 h at 37 °C. After washing with PBS, the cells were incubated with Hoechst 33258 (Sigma, St. Louis, MO, USA) and secondary antibodies (FITC-conjugated goat anti-mouse IgG (H + L), 1:50, or Cy3-conjugated goat anti-rabbit IgG (H + L), 1:100) for 1 h at 37 °C.

For Western blotting assays, freshly released parasites were purified by passing through 5 μm filters, centrifuged at 1000× *g* for 8 min, and washed with PBS. Purified parasites were lysed with RIPA buffer (Beyotime, Beijing, China) followed by SDS-PAGE. Subsequently, the separated proteins were transferred onto polyvinylidene fluoride (PVDF) membranes and blocked with 5% (*w/v*) skim milk for 1 h at room temperature. The primary antibody was incubated for 1 h at 37 °C. After washing with PBST (1% Tween-20), the membranes were incubated with horseradish peroxidase (HRP)-conjugated goat anti-mouse IgG (H + L) secondary antibody for 1 h at room temperature.

### 4.3. Construction of Transgenic Strains

To construct the *dsk2b* gene knockout strain and *dsk2a* and *dsk2b* double-gene knockout strain, we first constructed the pDHFR-*dsk2b* and pSAG1-CAS9-U6gRNA (*dsk2b*) plasmids. The pDHFR-*dsk2b* plasmid was composed of DHFR-TS cassette (amplified from the pDMG plasmid using primer 5 and primer 6), 707-bp 5′ sequences and 670-bp 3′ sequences of *dsk2b* (amplified from genomic sequence of RHΔ*ku80* strain using primer 1 and primer 2, primer 3 and primer 4 respectively) and the plasmid skeleton (amplified from pTCR-CD plasmid preserved in our laboratory using primer 7 and primer 8). These four fragments were linked by seamless cloning. The pSAG1-CAS9-U6gRNA (*dsk2b*) plasmid was constructed by cloning gRNA (targeting *dsk2b*) into the pSAG1-CAS9-U6gRNA (UPRT) by seamless cloning. The gRNA was designed on E-CRISP (http://www.e-crisp.org/ECRISP/, accessed on 10 November 2020). These two plasmids were transfected into RHΔ*ku80* or Δ*dsk2a* parasites followed by selection with pyrimethamine. All primer and sgRNA sequences were listed in Appendix A.

To determine the localization of DSK2b, 3× HA tags were fused to the N-terminal of the DSK2b coding sequence and cloned into a pDMG plasmid (pDMG-HA-DSK2b) [29]. The transcription of HA-DSK2b was initiated by *T. gondii* GRA1 promoter. Extracting the plasmid (non-linearized) and transfecting into the RHΔ*ku80* strain was followed by selection with pyrimethamine.

### 4.4. Invasion Assay

A total of 1 × 10^6^ freshly released tachyzoites of different strains were inoculated in confluent HFF cells growing on coverslips. After parasite invasion for 30 min, the cells were washed with PBS three times, fixed with 4% formaldehyde and subjected to IFA. The non-invaded (attached) tachyzoites were stained with rabbit anti-SAG1 before permeabilization, and then the invaded tachyzoites were stained with mouse anti-IMC1 after permeabilization (0.25% Triton X-100). FITC-conjugated goat anti-rabbit IgG (H + L) and Cy3-conjugated goat anti-mouse IgG (H + L) were secondary antibodies and were incubated together. The invaded tachyzoites were red-labeled and external tachyzoites were green-labeled. The invasion efficiency was calculated by counting the ratio of red-labeled/total tachyzoites in several random fields under a fluorescence microscope [30].

### 4.5. Proliferation Assay

A total of 4 × 10^5^ freshly released tachyzoites of different strains were inoculated in confluent HFF cells growing on coverslips. After parasite invasion for 30 min, the cells were washed with PBS, and the culture medium was replaced. After 24 h, the cells on the coverslips were fixed and subjected to IFA (an anti-GAP45 antibody was used to stain the parasites). Tachyzoites were counted in 100 parasitophorous vacuoles in several random fields visualized under a fluorescence microscope.

### 4.6. Plaque Assays

Plaque assays were performed as described previously [31]. Briefly, 500 freshly released tachyzoites were inoculated into confluent HFF monolayers growing in 6-well plates. After culture for 7 days, cells were washed with PBS and then fixed with 4% formaldehyde for 20 min, stained with 2% crystal violet for 10 min, washed with PBS, dried and imaged.

### 4.7. Virulence Assay in Mice

Five 6 week old female BALB/c mice were infected with 100 tachyzoites by intraperitoneal injection. The survival rate was monitored for 10 days post-infection.

### 4.8. Ethics Statement

Animal experiments were performed in accordance with the recommendations of the Guide for the Care and Use of Laboratory Animals of the Ministry of Science and Technology of China and the Institutional Animal Care and Use Committee of China Agricultural University (under the certificate of Beijing Laboratory Animal employee ID: CAU20161210–2, 10 December 2016).

### 4.9. Statistical Analysis

Graphs were generated and statistical analyses were performed using GraphPad Prism software v5.0 (GraphPad, San Diego, CA, USA). All data were analyzed using unpaired two-tailed Student’s *t* tests (Figure 2e, Figure 3d,e,h, Figure 4d and Figure 6c) or two-way ANOVA (Figure 3f and Figure 6a). The survival curves were statistically analyzed using the “Curve comparison” function in GraphPad Prism software.

## 5. Conclusions

*Toxoplasma gondii* is an important human and veterinary pathogen. The UPS regulates many important processes in *T. gondii*. UBL-UBA shuttle proteins are important components of the UPS. In this study, we identified a novel UBL-UBA shuttle protein in *T. gondii*. By constructing a gene knockout strain, we revealed its important role in the degradation of ubiquitinated proteins and the division of the parasite. In addition, autophagy occurred in the mutant strain. However, further functions of this novel UBL-UBA shuttle protein need to be explored.

## Figures and Tables

**Figure 1 ijms-22-07943-f001:**
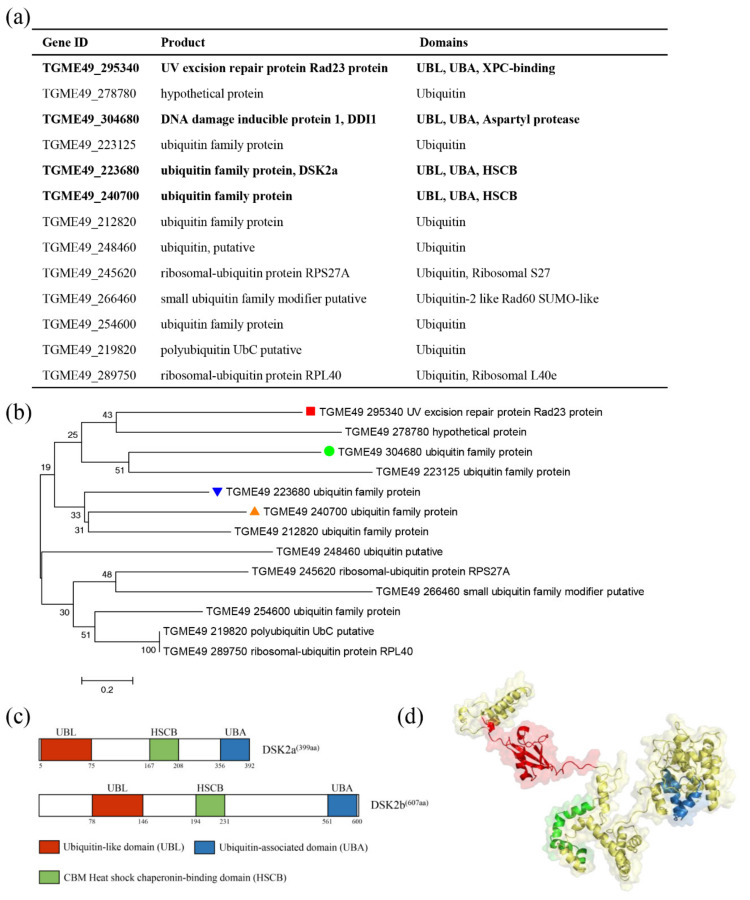
DSK2b is a novel UBL-UBA shuttle protein in *T. gondii.* (**a**) Proteins identified by BLAST in ToxoDB using ubiquitin as query sequence. Domains predicted by SMART database in these proteins are indicated. Bold font shows the proteins containing UBL and UBA domains. (**b**) Phylogenetic analysis of the identified proteins. The red block, green circle, blue triangle and orange triangle indicate RAD23, DDI1, DSK2a and DSK2b, respectively. (**c**) Domain architecture of DSK2a and DSK2b. UBL, ubiquitin-like domain; UBA, ubiquitin-associated domain; HSCB, CBM heat-shock chaperonin-binding domain. (**d**) The predicted three-dimensional (3D) structures of DSK2b. Several β sheets and an α-helix constitute the classical UBL domain (red); a bundle of three α-helices constitute the classical UBA domain (blue).

**Figure 2 ijms-22-07943-f002:**
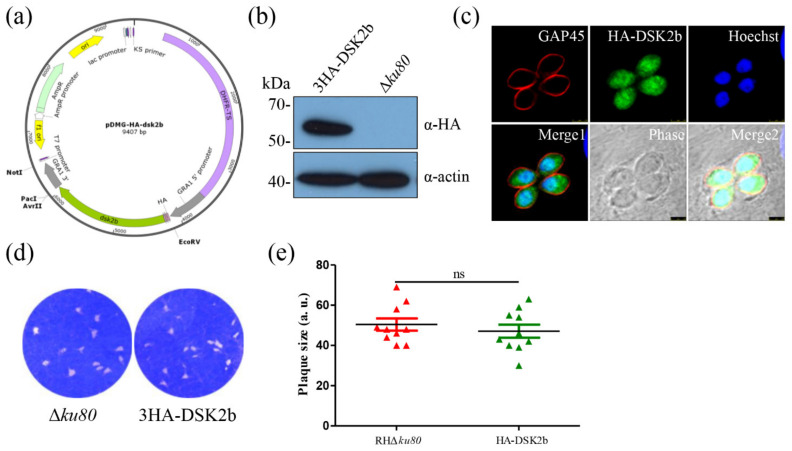
DSK2b is localized in the cytoplasm and nucleus of *T. gondii*. (**a**) Map of the pDMG-HA-DSK2b plasmid. The expression of HA-DSK2b was under the control of the *T. gondii* GRA1 promoter. (**b**) Western blotting identified the expression of HA-DSK2b. Lysates of the parasite were probed with anti-HA monoclonal antibody (upper panel). Actin served as the loading control (bottom panel). (**c**) Immunofluorescence assays determined the localization of HA-DSK2b. Parasites were labeled with mouse anti-HA monoclonal antibody (green) and rabbit anti-TgGAP45 (red). Hoechst was used to stain the nuclei (blue). Scale bar: 2.5 μm. (**d**,**e**) Plaque assay of the HA-DSK2b overexpression strain. A total of 500 tachyzoites were inoculated in HFFs for 7 days and then stained with crystal violet (**d**). Quantification of plaque sizes (**e**). 10 plaques were quantified; ns: not significant.

**Figure 3 ijms-22-07943-f003:**
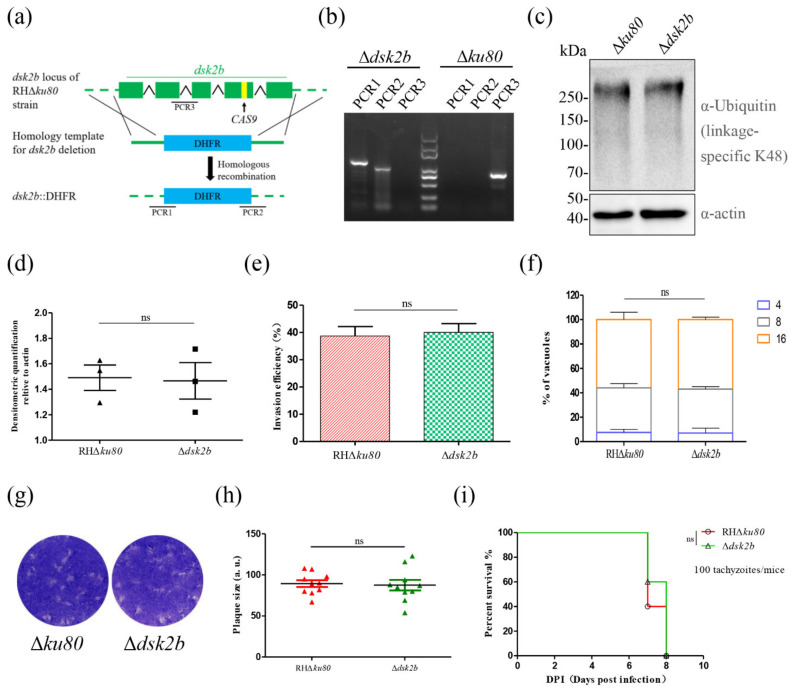
Deletion of DSK2b did not affect the degradation of ubiquitinated proteins and parasite growth. (**a**) Schematic representation of Δ*dsk2b* construction. The *dsk2b* gene locus in RHΔ*ku80* strain was replaced by DHFR cassette. (**b**) Diagnostic PCR of *dsk2b* deletion. The PCR1 and PCR2 suggest successfully homologous integration. PCR3 amplified the fragment of *dsk2b*. (**c**,**d**) Western blotting determined the ubiquitinated proteins in Δ*dsk2b* and Δ*ku80*. Ubiquitinated proteins were detected by rabbit anti-ubiquitin (linkage-specific K48) monoclonal antibody (upper panel). Actin served as a loading control (bottom panel). Graphs represent densitometric relative to actin (mean ± SD from three biologically independent experiments); ns: not significant. (**e**) Invasion assay of Δ*dsk2b* and Δ*ku80* strains. IFA was performed after invasion for 30 min. Tachyzoites and host cells were counted in several random fields under a fluorescence microscope, and the invasion efficiency was calculated. Graph represents mean ± SEM for three independent experiments. (**f**) Replication assay of Δ*dsk2b* and Δ*ku80* strains. IFA was performed after growth in HFF cells for 24 h. Anti-GAP45 antibody was used to stain parasites. Tachyzoites were counted in 100 PVs. Data represents mean ± SEM for two independent experiments. (**g**,**h**) Plaque formation of Δ*dsk2b* and Δ*ku80* strains. A total of 500 tachyzoites were inoculated in HFFs for 7 days and then stained with crystal violet (**g**). Quantification of plaque sizes (**h**). 10 plaques were quantified; ns: not significant. (**i**) Survival curve of BALB/c mice intraperitoneally injected with 100 tachyzoites of Δ*dsk2b* or Δ*ku80* strain for 8 days.

**Figure 4 ijms-22-07943-f004:**
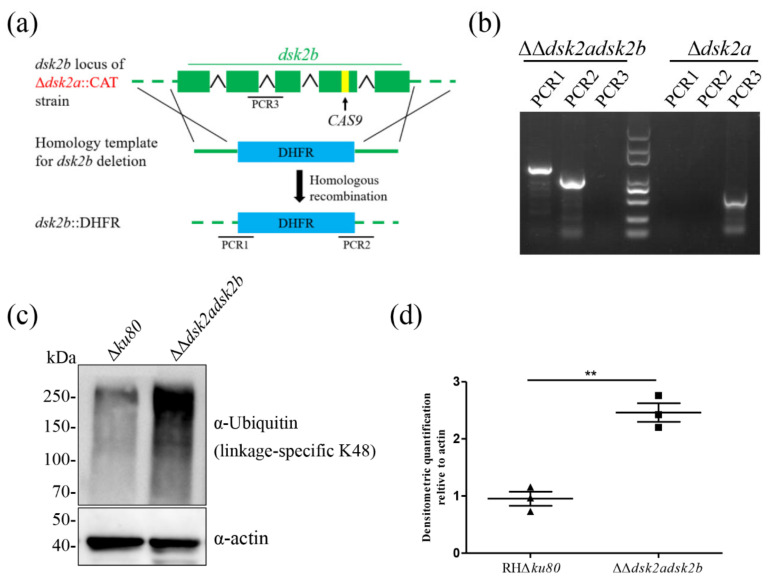
Deletion of DSK2s results in the accumulation of ubiquitinated proteins. (**a**) Schematic representation of ΔΔ*dsk2adsk2b* construction. The *dsk2b* gene locus in Δ*dsk2a::*CAT strain was replaced by DHFR cassette. (**b**) Diagnostic PCR of ΔΔ*dsk2adsk2b* construction. PCR1 and PCR2 were used to identify the homologous integration. PCR3 was used to amplify the fragment of *dsk2b*. (**c**,**d**) Western blotting determined the ubiquitinated proteins in ΔΔ*dsk2adsk2b* and Δ*ku80*. Ubiquitinated proteins were detected by rabbit anti-ubiquitin (linkage-specific K48) monoclonal antibody (upper panel). Actin served as a loading control (bottom panel). Graphs represent densitometric relative to actin. (mean ± SD from three biologically independent experiments). ** *p* < 0.01.

**Figure 5 ijms-22-07943-f005:**
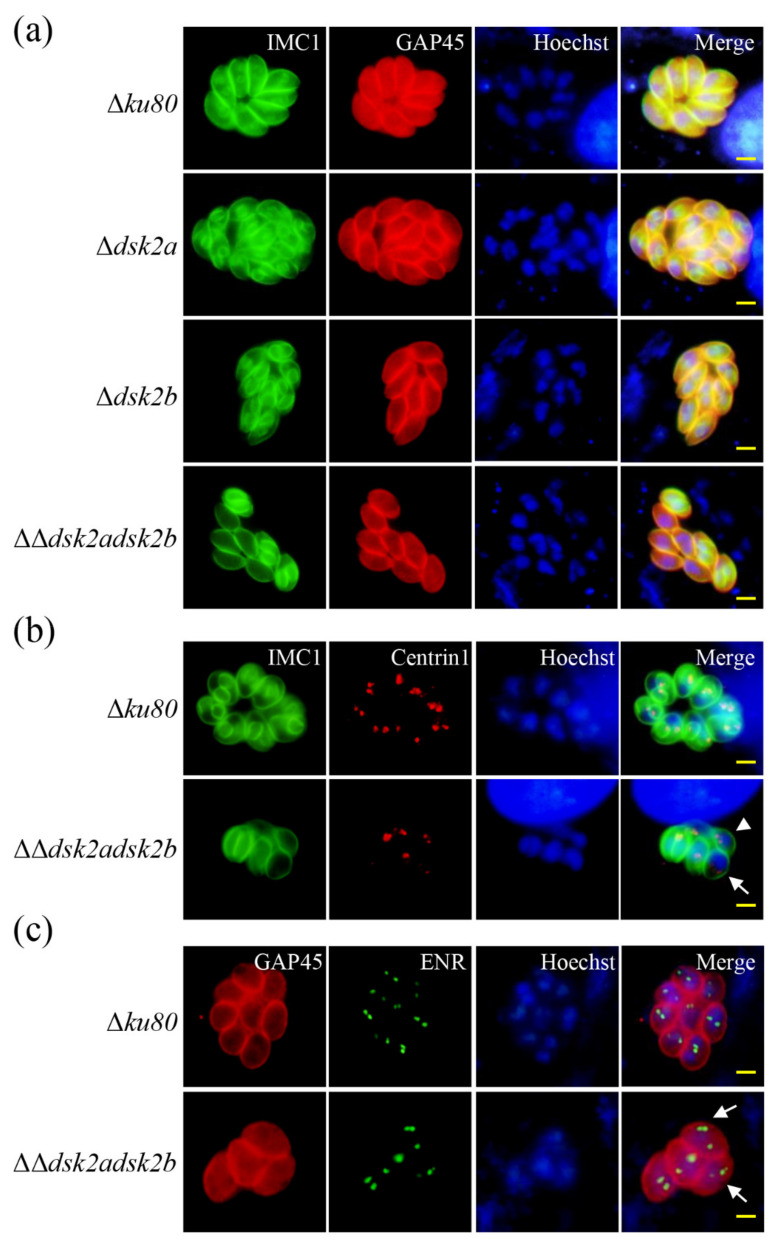
Deletion of DSK2s results in asynchronous replication of *T. gondii*. (**a**) IFA detected the division of Δ*ku80*, Δ*dsk2a*, Δ*dsk2b* and ΔΔ*dsk2adsk2b*. IMC1 was used to stain the daughter parasites. GAP45 was used to label the parental parasites. Scale bar: 2.5 μm. (**b**) IFA detected the duplication of a centrosome in Δ*ku80* and ΔΔ*dsk2adsk2b*. Centrin1 was used to label the centrosome. White triangle indicates the tachyzoite in which the duplication of centrosome occurs before daughter IMC formation. Arrow indicates the tachyzoite in which the duplication of centrosome is asynchronous. Scale bar: 2.5 μm. (**c**) IFA detected apicoplasts in Δ*ku80* and ΔΔ*dsk2adsk2b*. ENR was used to label apicoplasts. Arrows indicate tachyzoites with more than two apicoplasts. Scale bar: 2.5 μm.

**Figure 6 ijms-22-07943-f006:**
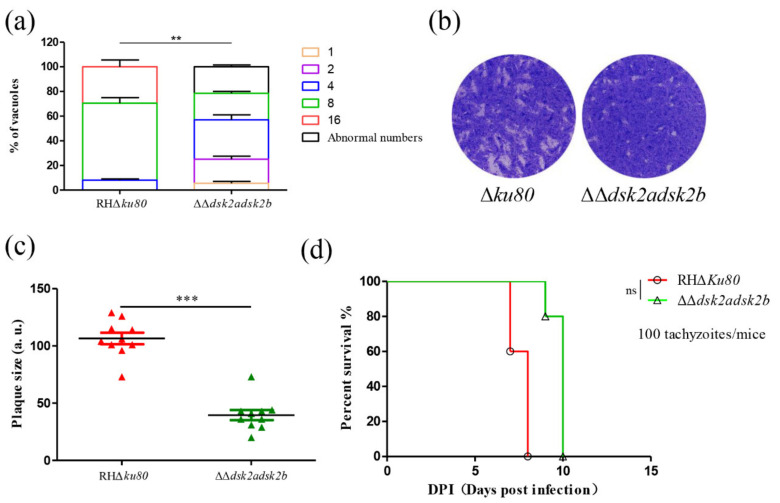
Deletion of DSK2s inhibits *T. gondii* growth. (**a**) Replication assay of the ΔΔ*dsk2adsk2b* and Δ*ku80* strains. IFA was performed after growth in HFF cells for 24 h. Anti-GAP45 antibody was used to stain parasites. Tachyzoites were counted in 100 PVs. Data represents means ± SEM for two independent experiments. ** *p* < 0.01. (**b**,**c**) Plaque formation of ΔΔ*dsk2adsk2b* and Δ*ku80* strains. A total of 500 tachyzoites were inoculated in HFFs for 7 days and then stained with crystal violet (**b**). Quantification of plaque sizes (**c**). 10 plaques were quantified. *** *p* < 0.001. (**d**) Survival curve of BALB/c mice intraperitoneally injected with 100 tachyzoites of ΔΔ*dsk2adsk2b* or Δ*ku80* strain for 10 days. ns: not significant.

**Figure 7 ijms-22-07943-f007:**
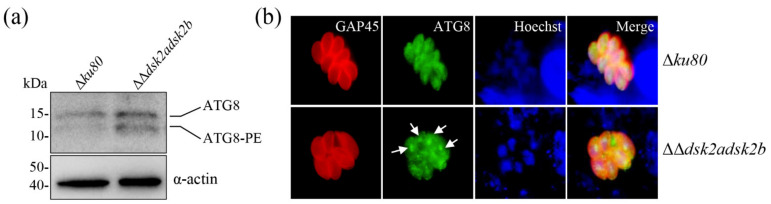
Autophagy pathway occurred in DSK2s mutant strain. (**a**) Western blotting detected ATG8 and ATG8-PE in ΔΔ*dsk2adsk2b* and Δ*ku80*. ATG8 was detected by mouse anti-ubiquitin polyclonal antibody (upper panel). Actin served as a loading control (bottom panel). (**b**) IFA detected ATG8 in ΔΔ*dsk2adsk2b* and Δ*ku80*. Parasites were labeled with mouse anti-ATG8 polyclonal antibody (green) and rabbit anti-TgGAP45 (red). Arrow indicates the aggregated ATG8. Hoechst was used to stain the nuclei (blue). Scale bar: 2.5 μm.

## Data Availability

Not applicable.

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
