# Peer review of "Toxoplasma gondii UBL-UBA Shuttle Protein DSK2s Are Important for Parasite Intracellular Replication"

_ijms, 2021, doi:10.3390/ijms22157943_

Round 1
Reviewer 1 Report
The manuscript "A novel UBL-UBA shuttle protein in Toxoplasma gondii participates in the degradation of ubiquitinated proteins" reports the characterization of a new protein in Toxoplasma gondii. This work is of great interest because it increases the understanding related to the biology of this pathogen. Experiments and results are well described, and the conclusions are supported by these results.
Comments:
- Manuscript elaboration. Some minor grammar and punctuation mistakes must be revised. One example: “To determine the localization of DSK2b in Toxoplasma, we firstly tagging the dsk2b gene with three epitope HA tags at the C-terminal of genomic locus, however, several attempts failed.” Replace tagging for the right time and separate the sentence after locus “…locus. However, …”.
- Line 41-42: Are you referring to authors from reference 7? Please, rephrase this sentence to clarify.
- Line 69: section title is placed wrongly.
- Lines 259-264: This information must be also mention in the results. Authors may explain about the strain clone and their phenotype. Then justify that in the discussion section.
- Line 293: Please, indicate the incubation time with primary antibodies.
- Figures quality must be improved. I imaging is because of the word file.
- Graphical abstract and conclusions section are missing. Please, include them.
After addressing the above-mentioned suggestions the work may be considered for publication on International Journal of Molecular Sciences.
Author Response
Response to Reviewer 1 Comments
- Manuscript elaboration. Some minor grammar and punctuation mistakes must be revised. One example: “To determine the localization of DSK2b in Toxoplasma, we firstly tagging the dsk2b gene with three epitope HA tags at the C-terminal of genomic locus, however, several attempts failed.” Replace tagging for the right time and separate the sentence after locus “…locus. However, …”.
Response: Thanks for your suggestion! We have corrected the mistakes in the revised manuscript (lines 99-101 and other place).
- Line 41-42: Are you referring to authors from reference 7? Please, rephrase this sentence to clarify.
Response: This sentence is misleading. We have rephrased this sentence in the revised manuscript (lines 43-45).
- Line 69: section title is placed wrongly.
Response: We have corrected this mistake in the revised manuscript (lines 73).
- Lines 259-264: This information must be also mention in the results. Authors may explain about the strain clone and their phenotype. Then justify that in the discussion section.
Response: Thanks for your suggestion! We have provided this information in the figure legends and emphasized the reproducible phenotype in different knockout clones in the discussion section (lines 273-276).
- Line 293: Please, indicate the incubation time with primary antibodies.
Response: We have indicated the incubation time in the revised manuscript (lines 324-325).
- Figures quality must be improved. I imaging is because of the word file.
Response: The figures were produced according to the requirements of the journal and inserted into the Word file. We have updated the figures in the revised manuscript, but we are not sure if it will be improved. We also think the figure quality would reduce in the Word file.
- Graphical abstract and conclusions section are missing. Please, include them.
Response: We have provided the conclusion section in the revised manuscript (section 5). However, it is hard to summarize a clearly graphical abstract according to the results in this manuscript. Of cause, if you insist, we will try to draw one. Thank you!
Reviewer 2 Report
Zhang et al identified a novel UBL-UBA shuttle protein DSK2b in Toxoplasma via bioinformatic analysis and studied its function on the degradation of ubiquitinated proteins. The authors did not observe any growth defects in parasite deficient in DSK2b, while deletion of both DSK2b and its homolog DSK2a caused asynchronous parasite replication and inhibited growth in vitro, but not attenuated virulence in mice in vivo. Although this is well performed study that reported the role of DSK2s in Toxoplasma intracellular replication, the details of methods and statistical analysis are poorly described, and the overall conclusions appear to be overreaching.
Major comment:
1. The current title does not match the data presented in the manuscript. Figure 3 clearly indicated that this novel UBL-UBA shuttle protein, DSK2b, individually plays no role in the degradation of ubiquitinated proteins. Figures 4, 5, and 6 together revealed that DSK2b synergizes with its homolog DSK2a to implement the UBL-UBA function inside the parasites. In addition, it lacks the details of how the authors detected the ubiquitinated proteins in Toxoplasma. I assume that the authors checked the ubiquitinated proteins in the extracellular parasites. Otherwise, it is hard for readers to know if these ubiquitinated proteins are not from the host cells. Nevertheless, I suggest the authors change the title to “Toxoplasma gondii UBL-UBA shuttle protein DSK2s are important for parasite intracellular replication” or similar.
2. The method section is overall poorly described and lacks details. For example:
- In the western blotting assay from section 4.2, the authors need to provide the information if the experiment was performed in extracellular parasites or intracellular parasites. If in extracellular parasites, how exactly the authors prepared such parasites.
- In section 4.3, the authors have to provide details of how they made those transgenic strains and the sequences of different primers/sgRNAs. Also, since the authors did ectopic expression in RHΔKu80 background (the strain has no NHEJ), please specify if the plasmid were linearized before transfection and if this is stably expression or not.
- In section 4.4, can the authors provide more details on how they distinguish the parasite invaded into the host cells with the parasite attached to the host cell surface?
- In section 4.9, the authors claimed that “all data were analyzed using unpaired two-tailed student’s t-test”. However, the parasites per vacuole data presented in Figures 3e and 6a needs to be analyzed by using two-way ANOVA. Please correct that.
- Please provide the number of independent biological replication for each experiment either in the method section or in the figure legends.
3. Figure 7a is very unconvincing. Please provide a clear image with a better resolution.
Minor comments and suggestions:
1. The authors sometimes use “gondii” while writing “Toxoplasma” at other times. Please unify this throughout the manuscript. If using “T.gondii”, please mention this is the abbreviation of “Toxoplasma gondii” when first use it.
2. It would be easier for readers to understand if the authors provide the schematic representation of Δdsk2b in Figure 3a.
3. The argument of not doing complementation due to the lack of selection markers in this strain is week. Other markers can still be used, for example, FUDR selection in UPRT-negative parasites, which the group has applied such strategy in several previous publications. I suggest the authors emphasize more on the reproducibility of parasite phenotype in different knockout clones. If the authors provided the data from the other three independent clones of the double knockout parasite as supplementary data, it would strengthen the argument.
Author Response
Response to Reviewer 2 Comments
Major comment:
- The current title does not match the data presented in the manuscript. Figure 3 clearly indicated that this novel UBL-UBA shuttle protein, DSK2b, individually plays no role in the degradation of ubiquitinated proteins. Figures 4, 5, and 6 together revealed that DSK2b synergizes with its homolog DSK2a to implement the UBL-UBA function inside the parasites. In addition, it lacks the details of how the authors detected the ubiquitinated proteins in Toxoplasma. I assume that the authors checked the ubiquitinated proteins in the extracellular parasites. Otherwise, it is hard for readers to know if these ubiquitinated proteins are not from the host cells. Nevertheless, I suggest the authors change the title to “Toxoplasma gondii UBL-UBA shuttle protein DSK2s are important for parasite intracellular replication” or similar.
Response: The current title dose not match the data indeed. Thus, we have changed the title as “Toxoplasma gondii UBL-UBA shuttle protein DSK2s are important for parasite intracellular replication”. Thanks for your suggestion!
The ubiquitinated proteins in the extracellular parasites were detected. We have provided the details in the revised manuscript (section 4.2).
- The method section is overall poorly described and lacks details. For example:
In the western blotting assay from section 4.2, the authors need to provide the information if the experiment was performed in extracellular parasites or intracellular parasites. If in extracellular parasites, how exactly the authors prepared such parasites.
Response: We have provided the details of the method in the revised manuscript (section 4.2).
In section 4.3, the authors have to provide details of how they made those transgenic strains and the sequences of different primers/sgRNAs.
Response: We have provided the details in the revised manuscript (section 4.3).
Also, since the authors did ectopic expression in RHΔKu80 background (the strain has no NHEJ), please specify if the plasmid were linearized before transfection and if this is stably expression or not.
Response: We did not linearize the plasmid before transfection (section 4.3). The expression of HA-DSK2b was detectable after several passages, suggesting its expression was stable rather than transient.
In section 4.4, can the authors provide more details on how they distinguish the parasite invaded into the host cells with the parasite attached to the host cell surface?
Response: We have provided the details in the revised manuscript (section 4.4).
In section 4.9, the authors claimed that “all data were analyzed using unpaired two-tailed student’s t-test”. However, the parasites per vacuole data presented in Figures 3e and 6a needs to be analyzed by using two-way ANOVA. Please correct that.
Response: We have corrected these mistakes (section 4.9), thank you!
Please provide the number of independent biological replication for each experiment either in the method section or in the figure legends.
Response: We have provided the information in the figure legends.
- Figure 7a is very unconvincing. Please provide a clear image with a better resolution.
Response: We have provided a new figure in the revised manuscript (Figure 7a).
Minor comments and suggestions:
- The authors sometimes use “gondii” while writing “Toxoplasma” at other times. Please unify this throughout the manuscript. If using “T.gondii”, please mention this is the abbreviation of “Toxoplasma gondii” when first use it.
Response: We have unified the word as “T. gondii” throughout the manuscript.
- It would be easier for readers to understand if the authors provide the schematic representation of Δdsk2b in Figure 3a.
Response: Thanks for your suggestion! We have provided the schematic representation of Δdsk2b construction (Figure 3a in the revised manuscript).
- The argument of not doing complementation due to the lack of selection markers in this strain is week. Other markers can still be used, for example, FUDR selection in UPRT-negative parasites, which the group has applied such strategy in several previous publications. I suggest the authors emphasize more on the reproducibility of parasite phenotype in different knockout clones. If the authors provided the data from the other three independent clones of the double knockout parasite as supplementary data, it would strengthen the argument.
Response: Thanks for your suggestions! We have rephrased this section to emphasize the reproducibility of phenotype in different knockout clones (lines 273-276).
Round 2
Reviewer 1 Report
Thank you to the authors for addressing all the suggestions. The additional information included in the revision adds quality and clarity to the manuscript. Related to figures quality, I can imagine that it is due to the Word format (they do not have good quality and it is hard to see details).
- Line 43: “These studies suggest that the UPS is exist in T. gondii and play important roles”. I would suggest to rephrase the sentence as: “These studies suggest that the UPS exists and plays important roles in T. gondii”
- When authors mentioned “for three independent experiments” (for example in figure 3 legend). Did they carry out the experiments in triplicate? Just to make sure; are in the replication assay (figure 3f) vacuoles with 4, 8 and 16 parasites? Did you not observed any 2 parasites vacuoles?
Author Response
Response to Reviewer 1 Comments
Thank you to the authors for addressing all the suggestions. The additional information included in the revision adds quality and clarity to the manuscript. Related to figures quality, I can imagine that it is due to the Word format (they do not have good quality and it is hard to see details).
- Line 43: “These studies suggest that the UPS is exist in T. gondii and play important roles”. I would suggest to rephrase the sentence as: “These studies suggest that the UPS exists and plays important roles in T. gondii”
Response: We have rephrased the sentence according to your suggestion (lines 41-42), thank you!
- When authors mentioned “for three independent experiments” (for example in figure 3 legend). Did they carry out the experiments in triplicate?
Response: Yes, the experiment was repeated three times.
Just to make sure; are in the replication assay (figure 3f) vacuoles with 4, 8 and 16 parasites? Did you not observed any 2 parasites vacuoles?
Response: After replication in host cells for 24 h, most of vacuoles contain 16 parasites. There are indeed 2 parasites vacuoles, but the ratio is very few, thus we did not count them. The ratio of 4 parasites vacuoles were already few in figure 3f.
Reviewer 2 Report
The authors have addressed all my comments and have sufficiently described the details in the method section. As a result, the manuscript is much improved, and their thoroughness is much appreciated.
Author Response
Response to Reviewer 2 Comments
The authors have addressed all my comments and have sufficiently described the details in the method section. As a result, the manuscript is much improved, and their thoroughness is much appreciated.
Response: Thank you! We greatly appreciate your time and efforts to improve our manuscript for publication.